# Rheological Properties of Wheat–Flaxseed Composite Flours Assessed by Mixolab and Their Relation to Quality Features

**DOI:** 10.3390/foods8080333

**Published:** 2019-08-09

**Authors:** Georgiana Gabriela Codină, Ana Maria Istrate, Ioan Gontariu, Silvia Mironeasa

**Affiliations:** Faculty of Food Engineering, Ştefan cel Mare University, 13 Universităţii Street, 720229 Suceava, România

**Keywords:** wheat–flaxseed composite, analytical quality, Mixolab, principal component analysis

## Abstract

The effect of adding brown and golden flaxseed variety flours (5%, 10%, 15% and 20% *w*/*w*) to wheat flours of different quality for bread-making on Mixolab dough rheological properties and bread quality was studied. The flaxseed–wheat composite flour parameters determined such as fat, protein (PR), ash and carbohydrates (CHS) increased by increasing the level of flaxseed whereas the moisture content (MC) decreased. The Falling Number values (FN) determined for the wheat–flaxseed composite flours increased by increasing the level of flaxseed. Within Mixolab data, greater differences were attributed to the eight parameters analysed: water absorption, dough development time, dough stability and all Mixolab torques during the heating and cooling stages. Also, a general decreased was also recorded for the differences between Mixolab torques which measures the starching speed (C3-2), the enzymatic degradation speed (C4-3) and the starch retrogradation rate (C5-4), whereas the difference which measures the speed of protein weakening due to heat (C1-2) increased. Composite dough behaviour presented a close positive relationship between MC and DT, and FN and PR with the C1-2 at a level of *p* < 0.05. The bread physical and sensory quality was improved up to a level of 10–15% flaxseed flour addition in wheat flour.

## 1. Introduction

Bread is one of the most consumed food product all over the world. However, the white bread obtained from refined wheat flour (WF) is rather high in carbohydrates and low in proteins, fibre, fat and minerals [1,2]. Therefore, nowadays the actual trend is to improve white bread quality from the nutritional point of view. The addition of flaxseed in bread-making may improve bread quality due to its composition because it is a rich source of essential amino acids, omega 3-fatty acid, dietary fibres, phenolic compounds, e.g., Tobias-Espinoza et al. [3], Oomah [4]. In the world there are many species of flaxseed (*Linum Usitatissimum* L.) varying in colour from brown to light gold [5,6]. The flaxseed varieties do not present significant differences in terms of their chemical composition, but only in terms of the amount of pigments present in the flaxseed, namely, the lower amount of pigments are, the lighter the seed colour is [7]. From the chemical composition point of view, the flaxseed contains about 40–50% fat content, 23–34% protein content, 4% mucilage and 5% ash [8]. It is the leading plant source in the alpha-linolenic acid content (omega-3 fatty acid), being five times higher than in canola oil and walnuts [5]. It is also a rich plant in some amino acids deficient in WF as lysine, valine and tryphtophan [9]. The flaxseed mucilage may be considered a food hydrocolloid due to its composition which consists of a mixture of neutral arabinoxylans and acidic rhamnose-containing polysaccharides [10]. Regarding its mineral content flaxseed is a rich source of K, Mg, Na, Cu, Mn, Zn and Fe [11]. 

Nowadays, two varieties of flaxseed are known all over the world, namely the golden and the brown one. Of the two varieties, the brown is more widely cultivated than the golden one. However, the golden variety is expected to minimally affect the colour of the final products. The effect of flaxseed flour addition on dough rheological properties was previously studied [12,13,14,15,16,17]. Some studies showed that dough stability decreased with the increased level of flaxseed addition [12,14] or may increase if high levels are incorporated in WF dough [15,16,17]. Dough extensibility decreased with the increased level of flaxseed flour addition [13]. Regarding dough behaviour during heating, very few studies have been made for wheat samples in which flaxseed flour was incorporated. However, it seems that flaxseed presents a delay effect on starch gelatinization process [12,13]. Regarding the effect of adding flaxseed to WF, studies have shown that bread quality was generally improved for the samples in which flaxseed flour was incorporated [15,18,19,20]. 

The aim of this study was to carry out a complex analysis of the effect of two varieties of flaxseed flour addition to refined, different quality wheat flours on dough rheological properties and bread quality. 

Though there have been previous reports on the physical and sensory characteristics of flaxseed-fortified bakery products, there has been a scarcity of reports on its effect on dough mixing and pasting behaviour using a complex device as the Mixolab and of studies between the physico-chemical parameters of wheat–flaxseed composite flours and rheological parameters of these ones. 

## 2. Materials and Methods

### 2.1. Flour Samples

Two commercial refined WFs with different qualities for bread-making were purchased from S.C. Mopan S.A. Company (Suceava, Romania). The samples were analysed according to the Romanian or international standard methods: gluten deformation and wet gluten according to SR 90/2007, moisture content according to ICC method 110/1, fat content according to ICC 136, protein content according to ICC 105/2, falling number according to ICC 107/1 and ash content according to ICC 104/1. The flaxseed varieties were purchased from SC DECO ITALIA SRL Cluj-Napoca, Romania and they were analysed for their chemical characteristics such as moisture content according to ICC 110/1, protein content according to ICC 105/2, fat content according to ICC 136 and ash content according to ICC 104/1. Carbohydrate content was determined as a difference of mean values: 100 − (the sum of the ash, protein, moisture content and fat) [21].

### 2.2. Flour Composites

Flours from two different flaxseed varieties (golden and brown) were incorporated in two commercial WFs at different levels (0, 5, 10, 15 and 20%) resulting in a set of 18 samples. The flour composites were mixed in ratios 100:0, 95:5, 90:10, 85:15 and 80:20 (*w*/*w*). For these purpose, two different quality WFs for bread-making were used. These were supplemented with each of the two types of flaxseed flour samples: the golden and the brown variety. The flour composites were analysed accordingly as follows; moisture content (ICC 110/1), fat content (ICC 136), protein content (105/2), ash content (ICC 104/1) and falling number (ICC 107/1). Carbohydrate content was determined as a difference of mean values: 100 − (the amount of the protein, moisture content, fat and ash) [21].

### 2.3. Evaluation of Flour Composite Dough Rheological Properties 

The dough mixing and pasting properties of the different wheat/flaxseed flour blends were studied using the Mixolab device (Chopin, Tripetteet Renaud, Paris, France). The composite flours rheological properties were determined according to ICC standard method No. 173. The Mixolab protocol was established as follows; total time to run the analysis 45 min, heating rate 4 °C/min and mixing temperature 30 °C. All the samples were made to optimum hydration level of composite flours in order to achieve the optimum consistency of dough corresponding to the C1 value of 1.1 N·m. The Mixolab parameters analysed were WA: water absorption (%); DT: dough development time (min); ST: dough stability (min); C2: minimum torque value, corresponding to the initial heating (N·m); C1-2: difference between C1 and C2 peak values (N·m), which measures the speed of protein weakening due to heat: C3, which expressed the starch gelatinization; C4, which expressed the stability of the starch gel formed; C5 torques (N·m), which expressed the starch retrogradation during the cooling stage and C3-2: the difference between C3 and C2 peak values (N·m), which measures the starching speed; C4-3: the difference between C4 and C3 peak values (N·m), which measures the enzymatic degradation speed; and C5-4: the difference between C5 and C4 peak values (N·m), which measures the starch retrogradation rate.

### 2.4. Bread-making

The bread formulations contained 100 g wheat–flaxseed composite flours (mixed in wheat: flaxseed ratios of 100:0, 95:5, 90:10, 85:15 and 80:20 (*w*/*w*)), commercial compressed yeast *Sacharomyces cerevisiae* type (3% flour basis), sodium chloride (2% flour basis) and water up to optimum wheat–flaxseed composite flour hydration capacity. All the ingredients were mixed at the speed of 200 rpm for 15 min in a laboratory mixer (Lancom, Shanghai, China). Then, the dough samples were modelled and placed into loaf pans. These were placed in a fermentation chamber (PL2008, Piron, Italy) for 60 min at 30 °C and 85% relative humidity. Finally, the samples were baked in an oven (PF8004D, Piron, Italy) for 30 min at 180 °C. Bread samples were cooled for 2 h, and then were subjected to physical and sensory analysis.

### 2.5. Evaluation of Bread Physical Characteristics

Bread physical characteristics (loaf volume, porosity, elasticity) were analysed according to the Romanian standard method SR 91:2007. 

### 2.6. Sensory Evaluation

The bread sensory characteristics were evaluated using semitrained panellists (20 persons). The overall acceptability, appearance, colour, flavour and texture of the samples were evaluated on a nine-point hedonic scale, scoring from one (extremely dislike) to nine (extremely like). 

### 2.7. Statistical Analysis

The statistical analysis of triplicate determinations was performed using the XLSTAT statistical package (free trial version 2016, Addinsoft, Inc., Brooklyn, NY, USA) at a significance level of *p* < 0.05. The data were analysed using variance analysis (ANOVA) and the Tukey test for mean comparison. Principal component analysis (PCA) was used to analyse the intercorrelation between all the variables studied using Statistical Package for Social Science (v.16, SPSS, Chicago, IL, USA). 

## 3. Results and Discussion

### 3.1. Flour Characteristics

The analytical characteristics of WF of two different qualities—strong quality (F1) and medium quality (F2)—for bread-making according to the Romanian standard SR 877:1996 used in this study are shown in Table 1. The falling number values of the both WFs shows that they have a low α amylase activity due to the fact that they have high FN values (>320 s) [22,23]. 

The chemical composition of brown and golden flaxseed, determined as a percentage of dried substance, is shown in Table 2. The moisture content of flaxseed samples ranged between 5.6% and 6.2%. Ash content of brown flaxseed was as high as 3.5% and that of golden flaxseed was of 3.41%. The fat content was in the range of 41.12 to 42.25% and protein content was between 19.74% and 20.85%, respectively.

The fat content of the flaxseed flours was in agreement with those reported by Ganorkar and Jain [24]. Carbohydrates in brown flaxseed were of 28.31% and of 29.02% for golden flaxseed variety.

### 3.2. Wheat–Flaxseed Composite Flours Physico-Chemical Characteristics

The physico-chemical characteristics of the wheat–flaxseed composite flours are shown in Table 3 and Table 4. 

In all cases, the increase in the level of flaxseed flour resulted in the in protein, fat, ash and falling number values and a decreased in moisture and carbohydrates content. This fact was expected due to the high flaxseed content in proteins, fats and ash and its lower amount of moisture and carbohydrates compared to WFs samples. A similar trend of these parameters values for the WF with different levels of flaxseed addition were also reported by Marpalle et al. [20], Codină et al. [13], and Wandersleben et al. [25]. The falling number increased by increasing the level of flaxseed addition more in the case of brown flaxseed variety than in that of the golden one.

It is well known that the falling number value is inversely correlated with α-amylase activity in flours [26] and therefore this trend shows that the flaxseed addition in WF decreased the α-amylase activity in the composite flours. 

### 3.3. Influence of Flaxseed on MixolabDough Rheological Properties

Incorporation of flaxseed from different varieties, brown and golden addition at 0, 5, 10, 15 and 20% in two categories of WF, namely strong (F1) and medium (F2), for bread-making, showed significant differences (*p*< 0.001) in terms of water absorption values. As it may be seen, the addition of brown flaxseed (BFs) and golden flaxseed (GFs), respectively in WF F1 and F2, decreased the water absorption (CH) values (Figure 1a). The lowest decreased in water absorption was found when the brown flaxseed was added in WF F2, from the level of 10% to 15% (55.2–54.6%). The decrease in water absorption in the case of golden flaxseed addition in F2 is in the same trend as the decrease in water absorption in F1. Similar effects on water absorption were observed by Roozegar et al. [16,17] and Codină et al. [12] when brown flaxseed or golden flaxseed were added, respectively. Kundu et al. [27] reported that the difference in water absorption is mainly caused by the gluten dilution, which needs less hydration, and therefore the wheat–flaxseed composite flours require lower amounts of water in the dough system in order to obtain the optimum consistency. As compared to the control samples (the sample without flaxseed flour addition), water absorption decreased in all the mixes made from wheat flour in which the golden or brown flaxseeds were incorporated. The lowest value for water absorption was obtained in the case when 20% level of brown flaxseed was incorporated in the flour of a medium quality for bread-making which decreased by 3.87% as compared to the control sample.

Compared to the wheat flour sample without flaxseed addition, highly significant effects (*p* < 0.001) were noticed in relation to dough development time (DT) values. The highest decreased in DT was observed for the sample with 20% flaxseed addition levels for both BFs and GFs varieties incorporated in F1 and F2, respectively (Figure 1b). These decreases may be due to gluten dilution in the dough system by flaxseed addition. Therefore, the amount of free water will increase, leading to a DT decrease due to the fact that in a dough system the highest amount of water is absorbed by starch and gluten [28] which will trigger a lower amount due to flaxseed addition in wheat flour.

Dough stability, which stands for dough strength, significantly decreased (Figure 1c) from 10.37 to 9.81 min and from 10.25 to 9.40 min when brown flaxseed and golden flaxseed were added in WF F1 at a level from 5% to 20%. A similar decrease in ST was noticed and in the case of F2 (from 10.87 to 9.73 min) when BFs was added and from 10.55 to 9.00 min when GFs was incorporated in WF, at the same levels of addition (5–20%). Similar results were reported by Pourabedin et al. [14], Meral and Dogan [15] and Roozegar et al. [16,17] for the addition of flaxseed in WF. However, as compared to the control sample, dough stability values gradually increased in mixes with the increase of levels addition from 0 to 10% for BFs and GFs, respectively, in WF F1, whereas for the WF F2 the ST values increased only for the levels of BFs and GFs up to 5% addition in wheat flour. This may likely be due to the interaction between polysaccharides (especially gums) and proteins in flaxseed–wheat composite flour as reported earlier by Rojas et al. [29]. Also, an increase of dough stability values to a higher level of flaxseed addition for the F1 flour than for the F2 one may be attributed to the WF quality. The F1 flour is of a strong quality for bread-making, which indicates that it can develop stronger and elastic dough than F2 flour which is of a medium quality. This indicates that F1 flour can sustain for a longer period of time higher wheat–flaxseed composite flour dough stability during mixing, compared to F2 flour. By comparing the obtained values for ST one can noticed that stability increased to a greater extent in the case of BFs addition in F1 from 8.02 min for the control sample (C) to 10.37 min for the sample with 5% BFs incorporation and from 8.57 min for the C to 10.87 min for the same level of BFs incorporation in F2. However, for a high level of flaxseed flour addition, a slight decrease in dough stability for both flours in which flaxseeds were incorporated due to gluten dilution was noticed because flaxseed is non-gluten flour.

The effect of incorporation of BFs and GFs flours at varying levels on dough C2 torque represents the protein weakening (C2) as illustrated in Figure 2a. Its values decreased with the increased level of flaxseed addition more in the case of brown variety than in the case when the golden one was used. The lowest values for C2 torque were recorded for the F2 flour with a decreased of 33.3% to a level of 20% brown flaxseed addition as compared to the control sample. 

A similar trend (Figure 2b) may be seen, and in the case of the difference between the peak C2 and C1 values (C2-1), which measures the speed of protein weakening due to heat, obviously increased with the increased level of flaxseed addition. An increase in the C2-1 values and a decrease in the value of C2 together with the increase in the flaxseed addition are due to the protein network structure. By flaxseed flour addition, proteins become less compact, a fact that favours the enzymatic attaching points, leading to an increase in the speed of protein weakening due to heat (C2-1) and a decrease in C2. Therefore, we may conclude according to the data obtained that by flaxseed addition the protein network becomes weaker under the effect of temperature increase.

When dough is heated above 60 °C the Mixolab begins to record the pasting properties of dough the C3 torque and the difference between the C3 and C2 peak values (C3-2) being associated with the starch gelatinization process. In general, both parameters values decreased (Figure 3a,b) with the increased level of flaxseed addition in the case of C3 values, this decrease is highly significant (*p* < 0.001). 

The decrease in C3 values is higher when the GF variety is added in WF dough with 26.2% for F1 and 14.4% for F2 for 20% level addition as compared to the control sample, probably due to starch dilution and the high content of fat and polysaccharides in flaxseed flour [12,14,29]. It is well known that the starch gelatinization process is influenced by the amylase-lipid complex formation, the amount of amylose leaching, the competition for free water between leached amylose and ungelatinised granules remained [30]. A decrease in C3 might be due to less swelling of the starch granules from WF in the presence of flaxseed flour of whose compounds may interact with amylose. Also, it is possible that some compounds from flaxseed to compete with starch to absorb water during the starch gelatinization process fact that will create difficulties for starch to gelatinize.

The C4 torque values corresponding to the hot starch stability paste decreased with the increased level of flaxseed addition (Figure 4a); more in the case of F2 with 17.14% when GFs were incorporated and with 13.1% when BFs were added in WF to a level of 20%. This effect may be attributed to the lower amylase activity in the wheat–flaxseed composite flour which slows gelatinization process and due to starch dilution, taking into account that flaxseed flour contains low amount of starch [12]. The amount of carbohydrates is less in the case of BFs than GFs and therefore the starch dilution of wheat–flaxseed composite flour is higher when GFs were added than in the case when BFs were incorporated in WF. Also, the stability of the starch gel formed is influenced by starch composition. Flaxseed flour addition in WF may have some interactions between starch and some compounds from the flaxseed flour. For example, the polysaccharides from the flaxseed content probably in a higher amount in BFs than in the case of GFs binds through the hydrogen bonds the water from the dough system, leading to a decrease of available water for the starch granules. Also, the high content of fat from the flaxseed flour may form insoluble complex with amylose leading to a decrease of the hot starch stability paste [31].

The difference between the C4 and C3 peak values (C4-3) did not vary in a significant way with the increased level of flaxseed addition. However, all the C4-3 presented lower values in the samples with flaxseed addition. This fact is explainable since the C4-3 values correspond to the rate of amylases hydrolysis on WF starch. Since the flaxseed flours did not bring amylases in dough system (as we can see from the falling number values) these parameters values decreased as compared to the sample when no flaxseeds were added. The starch retrogradation during the cooling period of the Mixolab device represented by the values of C5 torque and the difference between the C5 and C4 peaks (C5-4) decreased with the increased level of flaxseed addition (Figure 4b,c). This fact shows an anti-staling effect that flaxseed may have on bread quality more in the case of the golden variety than in the case of the brown one. An extent of bread freshness by flaxseed addition has also been reported by Khorshid et al. [19].

The decrease in the C5 values was higher when flaxseed was incorporated in F2 with 21.57% when GFs were added and 17.85% when BFs were added at a substitution level of 20% flaxseed in WF. This fact may be due to the high content of fat and other compounds in the flaxseed flour like polysaccharides that interact with gluten and starch in the dough system hindering a less starch retrogradation [14,19]. This fact is mainly attributed to the interaction between lipids from flaxseed flour with starch especially with amylose during the baking process. The complex formed between amylose and lipids is insoluble in water. In this form, amylose cannot leach out of starch granules. Thus, it decreased the amount of free amylase capable to leach out of gelatinized starch. As a consequence, it decreased its capacity to form intermolecular association during the cooling stage. On the other hand, the amylose remains inside the starch granules in a higher amount. This is due to the lipids presence, which creates difficulties between the associations of amylopectin molecules. Consequently, the starch retrogradation process is delayed [32]. Also, the polysaccharides from the flaxseed flours, which are probably in a lower amount in GFs than in BFs, may form intermolecular associations with leached amylose molecules during pasting which prevents starch retrogradation.

### 3.4. Correlation Analysis of the Evaluated Parameters for the Wheat–Flaxseed Composite Flours

The PCA was performed on wheat–flaxseed composite flours’ characteristics (moisture content (MC), fat content (Fat), protein content (PR), ash content (Ash), carbohydrates content (CHS), falling number (FN)) and dough rheological properties assessed by the Mixolab device (C2, C3, C4 and C5 torques; the difference between the C1 and C2 and peak values (C1-2); the difference between the C3 and C2 and peak values (C3-2); the difference between the C4 and C3 and peak values (C4-3); and the difference between the C5 and C4 and peak values (C5-4)) for the all 18 samples, which were analysed in this study, shown in Figure 5. The results obtained showed that all the variables used in order to perform PCA can be reduced to two principal components (PCs), 56.03% by PC1 and 26.37% by PC2. In PCA, factors extracted are retained if they have an eigenvalue >1 because they provide a lot more information than the initial variables, the first two components explaining 82.40% of the total variance. The plot of PC1 *vs.* PC2 plot shows along the PC1 axis, a close relationship between the starch pasting properties (C3, C3-2 and C4-3) and between wheat–flaxseed composite flours characteristics (FN, PR, Ash, Fat and MC and CH). It could be noticed that the Mixolab parameters related to dough rheological properties during heating (C1-2, C2, C3, C3-2, C4 and C4-3) were included in PC2 along with the starch retrogradation values at the cooling stage (C5 and C5-4). The influence of ST values was low, since their loadings for PC1 was close to null. Between wheat–flaxseed composite flours characteristics and Mixolab dough rheological properties a close relationship between MC and DT (*r* = 0.838) was noticed, respectively, C5 (*r* = 0.919) at a level of *p* < 0.01. FN was closely associated with the C1-2 value (*r* = 0.903) and inversely correlated to C2 (*r* = −0.910), C3-2 (*r* = −0.689), C3 (*r* = −0.808) at a level of *p* < 0.01. The positive correlation between FN and C1-2 may be due to the fact that the flours used in the analysis had a low α amylase activity and probably a low proteolytic activity. 

Since C1-2 expresses the speed of protein weakening due to heat, this speed is influenced by the proteolytic activity, and since the flaxseed addition did not improve α amylase activity, it is also probable that it did not to improve the proteolytic activity either. A negative correlation between FN and Mixolab values C3 and C3-2 is also explainable since both parameters are related to starch gelatinization process. The FN is a measure of the α amylase activity, its value being inversely correlated with the α amylase amount in WF and therefore high values of FN show low α amylase activity in dough system [33,34] fact that will negatively influence the starch gelatinization process. Protein content of the samples was significantly positively correlated with C1-2 value (r = 0.752) and negatively correlated with C2 (*r* = −0.754), C3-2 (*r* = −0.751), C3 (*r* = −0.812), C4-3 (*r* = −0.715) at a level of *p* < 0.01. Although flaxseeds have higher protein content, these are non-gluten, and therefore are weaker. Also, the Mixolab values related to dough pasting properties decreased due to the fact that by increasing the level of proteins in dough system the starch content decreased [35]. 

### 3.5. Influence of Flaxseed on Bread Physical Quality Characteristics

The physical characteristics of the bread samples with different flaxseed content are shown in Table 5 and Table 6. 

For all bread samples in which GFs were incorporated in the loaf volume, porosity and elasticity increased up to a level of 15% flaxseed addition and then decreased. Also, when BFs was added in WF, all the bread physical characteristics increased up to a level of 10% flaxseed addition and then decreased. An increase of the bread physical characteristics up to a level of 10–15% flaxseed addition may be due to the high amount of lipids from the flaxseed flour. This lipid effect on bread physical characteristics may be due to its presence in the liquid film which surrounds the gas cells. During baking, bread is not only gluten continuous but also gas continuous during the fact that gas cell opening occurs. This is one of the reasons why the bread does not collapse when gases are lost during baking and bread cooling [33]. The efficiency with which gas cell integrity was maintained is connected with the amount of lipids founds in the film which surrounds them and its semicrystalline organisation. The lipids are absorbed to the interface between gases and water forming a physical barrier in coalescence of the carbon dioxide bubbles. This leads to products with higher loaf volume, more fine and uniform porosity compared to the products without lipids addition. However, when high amount of flaxseed was added in WF, the bread physical characteristics begin to decrease, probably due to the gluten dilution effect. The flaxseed addition up to a certain level led to an increase in the bread physical characteristics. This was reported by [15,35,36]. 

As we can see from the Table 5 and Table 6 the physical values for the bread samples obtained from the medium quality flour are higher than the values for the bread obtained from the strong quality flour. This is probably due to the fact that F1 flour is a strong one and makes bread hardly extensible which will affects the growth of the bread. This will lead to products with low loaf volume. The F2 flour is of medium quality for bread-making with a good elasticity and extensibility. Also, it presents a high amount of gluten content which will facilitate a better holding for all dough components. This fact will improve the dough gas retention capacity during the bread-making process, which will lead to an increase of dough volume. As a consequence, the bread volume will increase as well [28]. 

### 3.6. Influence of Flaxseed on Bread Sensory Results

The results of sensory tests for the overall acceptability, appearance, colour, flavour and texture are shown in Figure 6 and Figure 7. 

The overall acceptability was concluded to be the best for bread samples in which flaxseed was incorporated up to 10%. No significant differences were found for samples with 5 and 10% BFs addition (*p*<0.05). Regarding the appearance and the colour evaluation, these parameters presented higher degree of liking for all bread samples up to a 10% GFs level. Samples with GFs received higher scores due to their more appealing, yellowish colour. The scores for flavour and texture were the highest for the samples in which 10% of BFs were added and for the samples in which 15% GFs were incorporated in WF. For all the sensory parameters evaluated significant differences were found between the control samples and the samples in which flaxseed flour was incorporated.

## 4. Conclusions

Physico-chemical and rheological properties of the composite flours varied significantly with the increased level of flaxseed addition. The partial substitution of wheat flour with flaxseed flours significantly increased (*p* < 0.001) the amounts of fats, proteins and carbohydrates to 9.69%, 13.84% and 62.99%, respectively, in the case of the wheat flour of a strong quality for bread-making, and to 9.53%, 14.13% and 62.34%, respectively, in the case of wheat flour of a medium quality for bread-making. It seems that flaxseed addition decreased the α-amylase activity in wheat flour since the falling number value increased with the increased level of flaxseed. The Mixolab results showed that water absorption, dough development time, protein weakening peak, starch gelatinization, starch hot-gel stability and retrogradation were significantly (*p* < 0.05) reduced as flaxseed level addition became higher. Stability was significantly (*p* < 0.05) increased up to a level of 5–10% flaxseed addition, after this level its values decreased but to a higher value than that of the control sample.

The graphic representation by PCA provides intuitive and quantitative classification of physico-chemical and Mixolab multidimensional data of wheat flours with different levels of flaxseed addition. The biplots presentation also shows good correlation between physico-chemical and rheological parameters measured by Mixolab device which greatly enhances the ability of understanding how the dough rheological properties provided by the Mixolab data can be affected by the physico-chemical parameters of composite flours. According to the bread quality parameters evaluated, a partial replacement of up to 10–15% flaxseed flour is possible in order to produce bread of a good quality.

## Figures and Tables

**Figure 1 foods-08-00333-f001:**
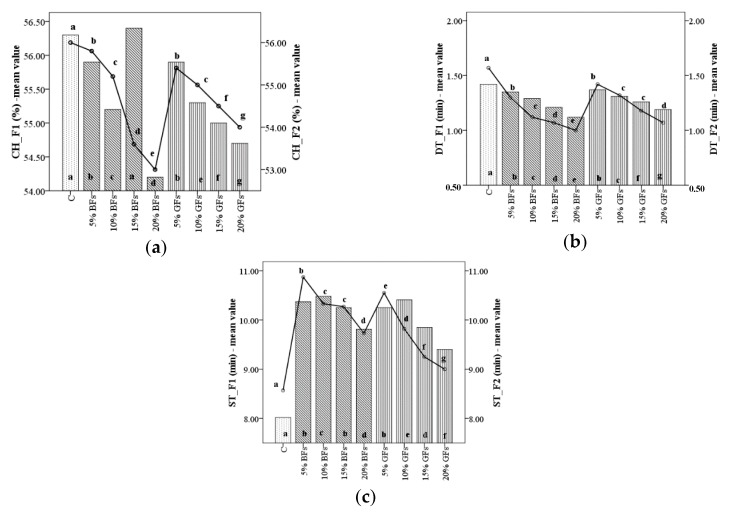
Mixolab parameters during mixing of WF (C) and flaxseed–wheat composite flours formulated by adding different levels (5,10,15 and 20%) of brown flaxseed (BFs) and golden flaxseed flour (GFs), respectively, in two types of flour—F1 and F2: (**a**) CH: water absorption; (**b**) DT: development time; and (**c**) ST: stability. Different letters indicate significant differences (*p* < 0.05) between samples.

**Figure 2 foods-08-00333-f002:**
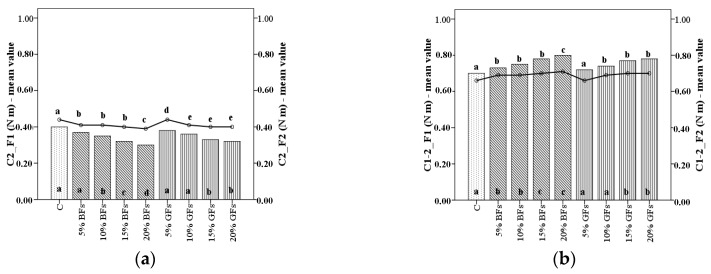
C2 torque (**a**) and difference between torques C1 and C2 (C1-2) (**b**) of WF (C) and flaxseed–wheat composite flours formulated by adding different levels (5%,10%,15% and 20%) of brown flaxseed (BFs) and golden flaxseed flour (GFs) in two types of flour—F1 and F2. Different letters indicate significant differences (*p* < 0.05) between samples.

**Figure 3 foods-08-00333-f003:**
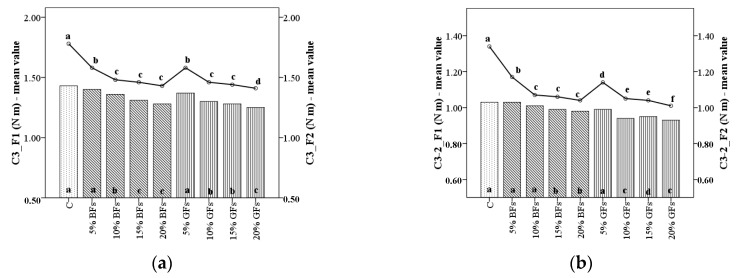
C3 torque (**a**) and difference between torques C3 and C2 (C3-2) (**b**) of WF (C) and flaxseed–wheat composite flours formulated by adding different levels (5%,10%,15% and 20%) of brown flaxseed (BFs) and golden flaxseed flour (GFs) in two types of flour—F1 and F2. Different letters indicate significant differences (*p* < 0.05) between samples.

**Figure 4 foods-08-00333-f004:**
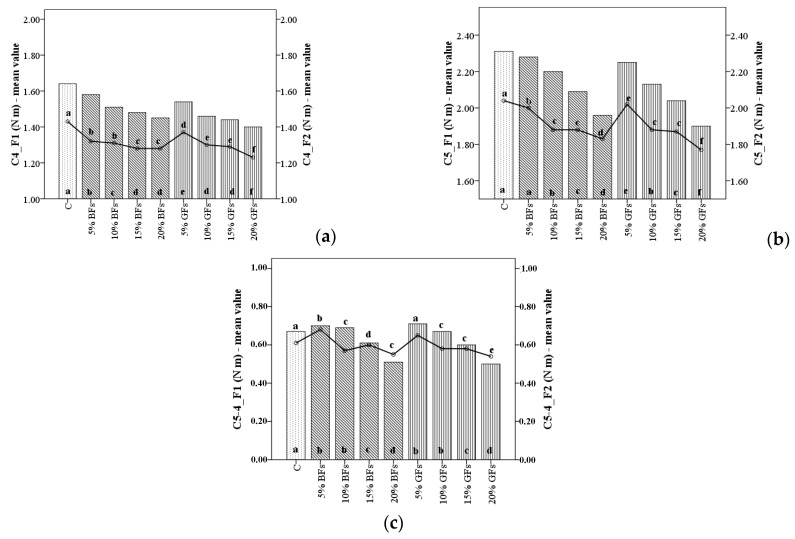
C4 torque (**a**), C5 torque (**b**) and difference between torques C5 and C4 (C5-4) (**c**) of WF (C) and flaxseed–wheat composite flours formulated by adding different levels (5%,10%,15% and 20%) of brown flaxseed (BFs) and golden flaxseed flour (GFs) in two types of flour, F1 and F2. Different letters indicate significant differences (*p* < 0.05) between samples.

**Figure 5 foods-08-00333-f005:**
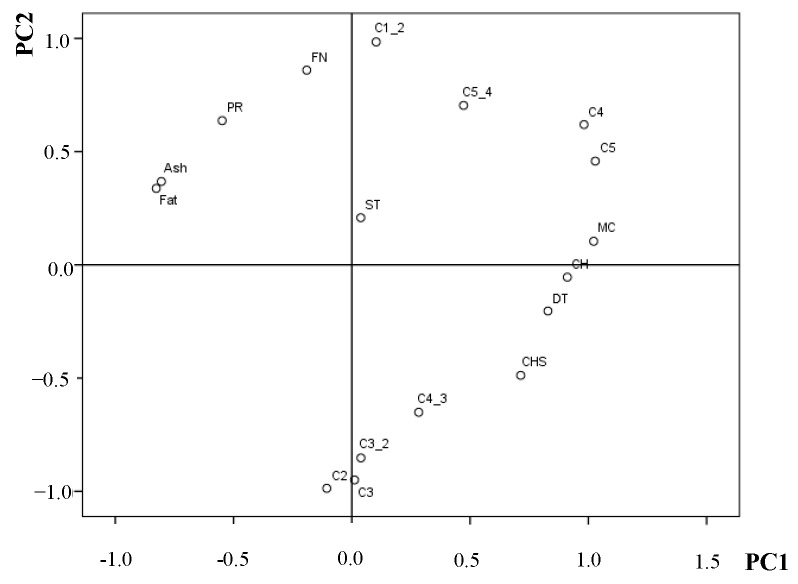
Loading plot of the first two principal components based on physicochemical and rheological properties of the wheat–flaxseed composite flours samples: MC: moisture content; Fat: fat content; PR: protein content; Ash: ash content; CHS: carbohydrates content; WA: water absorption; DT: development time; ST: stability; C2, C3, C4, and C5—Mixolab torques, C1-2, C3-2, C4-3, and C5-4 difference between Mixolab peak values C1 and C2, C3 and C2, C4 and C3 and C5 and C4.

**Figure 6 foods-08-00333-f006:**
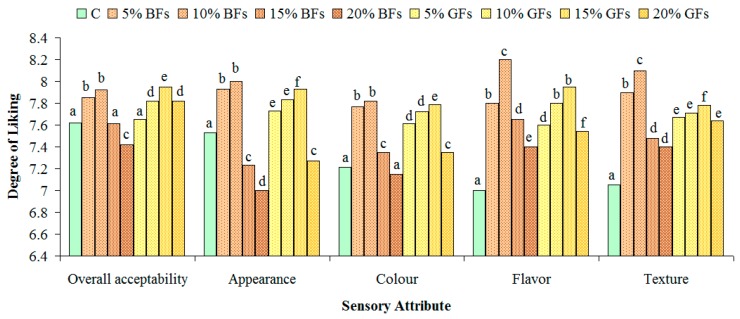
Effect of brown flaxseed (BFs) and golden flaxseed (GFs) at different addition levels: 0% (C), 5%, 10%, 15% and 20% on the sensory attributes of bread from a WF of a strong quality for bread-making. Means with different letters indicate significant difference among treatments (*p* < 0.05).

**Figure 7 foods-08-00333-f007:**
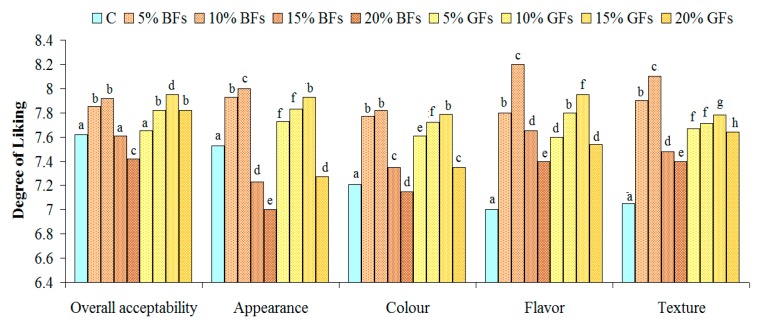
Effect of brown flaxseed (BFs) and golden flaxseed (GFs) at different addition levels: 0% (C), 5%, 10%, 15% and 20% on the sensory attributes of bread from a WF of a good quality for bread-making. Means with different superscripts indicate significant difference among treatments (*p* < 0.05).

**Table 1 foods-08-00333-t001:** Quality characteristics of wheat flour (mean value ± standard deviation).

Parameters	Strong Flour	Medium Flour
	F1	F2
Moisture (%)	13.90 ± 0.01	14.50 ± 0.01
Fat (%)	1.7 ± 0.01	1.5 ± 0.01
Protein content (%)	12.2 ± 0.01	12.6± 0.01
Ash content (%)	0.65 ± 0.01	0.65 ± 0.01
Carbohydrates (%)	71.55 ± 0.01	70.75 ± 0.01
Wet gluten (%)	27.50 ± 0.10	34.00 ± 0.20
Gluten deformation index (mm)	3.00 ± 0.18	8.00 ± 0.6
Falling Number index (s)	325.00 ± 2.65	380.00 ± 3.92

**Table 2 foods-08-00333-t002:** Parameters of brown and golden flaxseed flours.

Parameters	Mean Value ± Standard Deviation	
	Brown Flaxseed	Golden Flaxseed
Moisture content (%)	6.2 ± 0.07	5.6 ± 0.04
Fat (%)	42.25 ± 1.15	41.12 ± 1.03
Protein content (%)	19.74 ± 0.46	20.85 ± 0.42
Ash (%)	3.50 ± 0.03	3.41 ± 0.02
Carbohydrates (%)	28.31 ± 0.02	29.02 ± 0.01

**Table 3 foods-08-00333-t003:** The analysis of the variance of brown (BFs) and golden (GFs) flaxseeds addition to the strong quality flour for bread-making.

Characteristics	Type of Flaxseeds		*F* Ratio	Flaxseeds Doses (%)					*F* Ratio	Flaxseed Type × Doses
	BFs	GFs		0	5	10	15	20		
MC	13.12 ± 0.56 ^a^	13.06 ± 0.60 ^b^	13 **	13.90 ± 0.08 ^eab^	13.49 ± 0.018 ^dab^	13.10 ± 0.03 ^cab^	12.69 ± 0.05 ^ba^	12.30 ± 0.06 ^ab^	1154 ***	2 ^ns^
Fat	5.74 ± 2.96 ^a^	5.64 ± 2.88 ^b^	87 ***	1.70 ± 0.02 ^g^	3.65 ± 0.03 ^f^	5.68 ± 0.05 ^e^	7.69 ± 0.09 ^d^	9.69 ± 0.12 ^c^	62,448 ***	12 ***
PR	12.93 ± 0.59 ^a^	13.07 ± 0.64 ^b^	4.4 *	12.22 ± 0.02 ^eab^	12.58 ± 0.02 ^dab^	12.96 ± 0.36 ^cab^	13.41 ± 0.09 ^ba^	13.84 ± 0.16 ^ab^	74.5 ***	0.4 ^ns^
Ash	0.93 ± 0.20 ^a^	0.92 ± 0.20 ^a^	1.51 ^ns^	0.65 ± 0.00 ^eab^	0.78 ± 0.02 ^dab^	0.92 ± 0.02 ^cab^	1.06 ± 0.02 ^ba^	1.21 ± 0.02 ^ab^	466.38 ***	0.24 ^ns^
CHS	67.24 ± 3.29 ^a^	67.35 ± 3.10 ^a^	0.2 ^ns^	71.55 ± 0.02 ^eab^	69.60 ± 0.37 ^dab^	67.30 ± 0.67 ^cab^	64.98 ± 0.45 ^ba^	62.99 ± 0.07 ^ab^	398.1 ***	0.6 ^ns^
FN	385.40 ± 40.39 ^a^	360.93 ± 32.38 ^b^	502.6 ***	325.00 ± 1.78 ^g^	346.50 ± 17.52 ^f^	374.33 ± 18.40 ^e^	398.000 ± 15.44 ^d^	410.50 ± 29.08 ^c^	1013.2 ***	32 ***

The means values ± standard deviation in one row followed by different letters differ significantly different at * *p* < 0.05; ** *p* < 0.01; *** *p* < 0.001; ns: nonsignificantly (*p* > 0.05). MC: moisture content (%); Fat: fat content (%); PR: protein content (%); Ash: ash content (%); CHS: carbohydrate content (%); FN: falling number value (s); BFs: the mean values of WFs samples with different doses of brown flaxseed flours addition; GFs: the mean values of WFs samples with different doses of golden flaxseed flours addition.

**Table 4 foods-08-00333-t004:** The analysis of the variance of brown (BFs) and golden (GFs) flaxseeds addition to the medium quality flour for bread-making.

Characteristics	Type of Flaxseed		*F* Ratio	Flaxseeds Doses					*F* ratio	Flaxseed Type × Doses
	BFs	GFs		0	5	10	15	20		
MC	13.66 ±0.60 ^a^	13.60 ± 0.65 ^b^	25 **	14.50 ± 0.008 ^g^	14.06 ± 0.018 ^f^	13.64 ± 0.03 ^e^	13.20 ± 0.05 ^d^	12.78 ± 0.06 ^c^	2545 **	3 *
Fat	5.56 ± 2.98 ^a^	5.46 ± 2.89 ^b^	158 **	1.5 ± 0.008 ^g^	3.50 ± 0.03 ^f^	5.51 ± 0.06 ^e^	7.51 ± 0.09 ^d^	9.53 ± 0.13 ^c^	101,264 **	22 **
PR	13.30 ± 0.51 ^a^	13.43 ± 0.59 ^b^	48 **	12.60 ± 0.02 ^g^	13.03 ± 0.13 ^f^	13.36 ± 0.06 ^e^	13.74 ± 0.09 ^d^	14.13 ± 0.13 ^c^	728 **	4 *
Ash	0.93 ± 0.20 ^a^	0.92 ± 0.20 ^a^	1.41 ^ns^	0.65 ± 0.008 ^ga^	0.78 ± 0.01 ^fa^	0.92 ± 0.01 ^ea^	1.06 ± 0.01 ^da^	1.21 ± 0.02 ^ca^	871.19 **	0.31 ^ns^
CHS	66.52 ± 3.10 ^a^	66.59 ± 3.05 ^b^	79 **	70.73 ± 0.01 ^g^	68.67 ± 0.02 ^f^	66.55 ± 0.04 ^e^	64.46 ± 0.06 ^c^	62.34 ± 0.07 ^c^	148,546 **	11 **
FN	440.06 ± 20.26 ^a^	412.00 ± 29.60 ^b^	1738 **	380.33 ± 2.25 ^g^	391.83 ± 5.87 ^f^	420.00 ± 16.49 ^e^	454.50 ± 25.75 ^d^	483.50 ± 29.08 ^c^	3268 **	230 **

The means values ± standard deviation in one row followed by different letters differ significantly different at * *p* < 0.05; ** *p* < 0.001; ns: nonsignificantly (*p* > 0.05). MC: moisture content (%); Fat: fat content (%); PR: protein content (%); Ash: ash content (%); CHS: carbohydrate content (%); FN: falling number value (s); BFs: the mean values of WFs samples with different doses of brown flaxseed flours addition; GFs: the mean values of WFs samples with different doses of golden flaxseed flour addition.

**Table 5 foods-08-00333-t005:** Analysis of variance of the influence of flaxseeds from the brown variety (BFs) and golden variety (GFs) addition on physical characteristics of bread obtained from strong quality flour for bread-making.

Characteristics	Type of Flaxseed		*F* Ratio	Flaxseeds Doses (%)					*F-*Ratio	Flaxseed Type × Doses
	BFs	GFs		0	5	10	15	20		
Loaf specific volume (cm^3^/100 g)	290.97 ± 27.56 ^a^	294.11 ± 13.80 ^b^	74 **	277.49 ± 0.89 ^e^	295.44 ± 14.59 ^d^	317.61 ± 16.24 ^c^	296.96 ± 18.56 ^b^	275.22 ± 20.84 ^a^	1774 **	1548 **
Porosity (%)	84.30 ± 1.16 ^a^	85.84 ± 2.67 ^b^	17.8 **	83.50 ± 0.89 ^e^	84.90 ± 1.17 ^d^	86.50 ± 1.25 ^c^	87.05 ± 3.03 ^b^	83.40 ± 0.95 ^a^	16.9 **	7.9 **
Elasticity (%)	85.00 ± 1.09 ^a^	86.06 ± 2.31 ^b^	8.4 *	84.20 ± 0.90 ^e^	85.25 ± 0.97 ^d^	86.10 ± 0.92 ^c^	87.77 ± 2.49 ^b^	84.32 ± 0.89 ^a^	13.0 **	4.9 *

The means values ± standard deviation in one row followed by different letters differ significantly different at * *p* < 0.01; ** *p* < 0.001.

**Table 6 foods-08-00333-t006:** Analysis of variance of the influence of flaxseeds from the brown variety (BFs) and golden variety (GFs) addition on the physical characteristics of bread obtained from a medium quality flour for bread-making.

Characteristics	Type of Flaxseed		*F* Ratio	Flaxseeds Doses (%)					*F*-Ratio	Flaxseeds Type × Doses
	BFs	GFS		0	5	10	15	20		
Loaf specific volume (cm^3^/100 g)	351.19 ± 18.34 ^a^	373.58 ± 44.24 ^b^	3760 *	332.15 ± 0.89 ^e^	347.84 ± 8.62 ^d^	389.41 ± 10.10 ^c^	403.80 ± 45.09 ^b^	338.73 ± 2.57 ^a^	6196 *	1827 *
Porosity (%)	84.30 ± 2.19 ^a^	85.04 ± 1.81 ^a^	4.1 ^ns^	83.50 ± 0.89 ^e^	85.63 ± 0.89 ^d^	87.11 ± 0.91 ^c^	84.97 ± 1.19 ^b^	82.13 ± 1.38 ^a^	22.2 **	1.2 ^ns^
Elasticity (%)	84.99 ± 1.83 ^a^	85.66 ± 1.65 ^a^	3.3 ^ns^	84.20 ± 0.90 ^e^	85.95 ± 0.97 ^d^	86.71 ± 1.00 ^c^	86.30 ± 1.96 ^b^	83.48 ± 1.26 ^a^	11.9 **	4.5 *

The means values ± standard deviation in one row followed by different letters differ significantly different at * *p* < 0.01; ** *p* < 0.001; ns: nonsignificantly (*p* > 0.05).

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
