# Peer review of "Rheological Properties of Wheat–Flaxseed Composite Flours Assessed by Mixolab and Their Relation to Quality Features"

_foods, 2019, doi:10.3390/foods8080333_

Round 1

Reviewer 1 Report

Reviewer’s comments:

This is an interesting article describing the rheological properties of two wheat flours enriched with flaxseed flour from two different varieties. There are however several points that need to be improved/clarified.

The use of English language should be revised across the manuscript and certain phrases should be re-written.

In the abstract it is preferable to mention the parameters (periphrastically) assessed by the Mixolab instrument instead of the C# so as the reader to be able to fully understand their meaning and the abstract to be a stand-alone text.

Introduction should not contain results and a short explanation on why/how the certain flours (wheat and flaxseed) were chosen for this study should be included.

Materials and methods:

It is not clear why all wheat and flaxseed flours were not initially analyzed for the same properties/characteristics (tables 1 and 2) using the standard methods described in paragraph 2.1. Please explain. Also, the protocol followed during Mixolab analyses should be described within the manuscript as well.

Results and discussion:

Results in many cases are foreseeable, while in several cases the explanations provided for the variance between different samples are general and trivial. Comments have been included within the manuscript in these points. Therefore, please check them and make appropriate corrections. PCA analysis should be checked as well.

General comments:

It would be preferable for a strong publication to also include breadmaking experiments as the 15-20% substitution of wheat with flaxseed could lead to products with significantly different quality and sensory characteristics. This factor should be taken into account as the rheological properties of the dough may improve in some cases but would the occurring bread still taste good? What modifications are necessary during breadmaking while substituting wheat with flaxseed? Also, a clear justification on how the findings of this research can contribute to the use of flaxseed flour in breadmaking should be preferably be included in the conclusions.       

More detailed comments can be found in the uploaded manuscript.

Reviewer 2 Report

The study is well presented althugh contains a lack of information that could improve the content a overall the interest of this topic. Rheological aspects are important to decide the use of different products, however the paper can based in the healthy benefits of this product and characterise only the tecnofunctional properties.
I think the paper could be considered if the autor study any other marker to correlate with the analysis obtained using the intruments Mixlab.
Although the tables contains the statistic the graph did not have this info so it would be necessary to incluide also in future manuscripts.

I suggest to the authors to incorpórate enzymatic or bioactive markers to correlate and improve results obtained with the rheometer.

Round 2

Reviewer 1 Report

The manuscript has been significantly improved with the addition of the quality and sensory characteristics of wheat-flaxseed bakery products. The work seems now more complete and well-built. Justifications have been included in several points in which results were raising concerns. However, I would recommend to read through the manuscript once again and edit English language as there as still several errors and some points can be described in a more simple and comprehensive way. More comments can be found in the uploaded manuscript.

Reviewer 2 Report

The paper has been improved and can be ocnsidered in actual format for publication.
